# Development of Smart Films of a Chitosan Base and *Robusta* Coffee Peel Extract for Monitoring the Fermentation Process of Pickles

**DOI:** 10.3390/foods12122337

**Published:** 2023-06-10

**Authors:** Jiatong Yan, Hongda Yu, Zhouhao Yang, Lin Li, Yuyue Qin, Haiyan Chen

**Affiliations:** 1Faculty of Food Science and Engineering, Kunming University of Science and Technology, Kunming 650550, Chinaseacome@163.com (H.C.); 2School of Life Healthy and Technology, Dongguan University of Technology, Dongguan 523830, China

**Keywords:** anthocyanins, color changes, *Robusta* coffee peel, smart packaging, monitor fermentation

## Abstract

Smart film is widely used in the field of food packaging. The smart film was prepared by adding anthocyanin-rich *Robusta* coffee peel (RCP) extract into a chitosan (CS)–glycerol (GL) matrix by a solution-casting method. By changing the content of RCP (0, 10%, 15% and 20%) in the CS–GL film, the related performance indicators of CS–GL–RCP films were studied. The results showed that the CS–GL–RCP films had excellent mechanical properties, and CS–GL–RCP15 film maintained the tensile strength (TS) of 16.69 MPa and an elongation-at-break (EAB) of 18.68% with RCP extract. CS–GL–RCP films had the best UV-vis light barrier property at 200–350 nm and the UV transmittance was close to 0. The microstructure observation results showed that CS–GL–RCP films had a dense and uniform cross section, which proved that the RCP extract had good compatibility with the polymer. In addition, the CS–GL–RCP15 film was pH-sensitive and could exhibit different color changes with different pH solutions. So, the CS–GL–RCP15 film was used to detect the fermentation process of pickles at 20 ± 1 °C for 15 days. The pickles were stored in a round pickle container after the boiling water had cooled. The color of the CS–GL–RCP15 film changed significantly, which was consistent with the change of pickles from fresh to mature. The color of the smart film changed significantly with the maturity of pickles, and the difference of Δ*E* of film increased to 8.89 (15 Days), which can be seen by the naked eye. Therefore, CS–GL–RCP films prepared in this study provided a new strategy for the development of smart packaging materials.

## 1. Introduction

Food has a low shelf life and is easily affected by endogenous enzymes and external microorganisms. In life, people’s awareness of food safety is increasing year by year. With the increasing demand for food safety, protecting consumers from diseases caused by spoiled food and maximizing food quality have become important research directions. Therefore, people have a strong interest in advanced packaging materials. Smart packaging has risen rapidly and has entered the sight of consumers. Smart food packaging can provide consumers with information, display changes in the food packaging environment and monitor the freshness of food in real time. It is of great significance to improve food safety and reduce food waste [1]. 

Smart packaging film is usually composed of film matrix material and a pH indicator. Various biopolymers have been applied to the preparation of films [2]. Chitosan has many various excellent performances, such as film-forming properties, biodegradability and bacteriostasis, and is widely used in the preparation of food packaging film [3]. However, the low mechanical strength of CS film limits its better utilization [4]. Glycerin is a food-grade plasticizer that does not cause consumer safety problems and can improve the mechanical strength of CS films [5]. Therefore, in this experiment, glycerin was added to CS film to meet the needs of food packaging [6].

With the progress of science and the development of recent technologies, the safety of a pH indicator has attracted more and more attention [7]. The smart packaging film often uses chemical synthetic dyes as a pH indicator, such as chlorophenol red, bromocresol purple and bromophenol blue. However, chemical dyes have certain toxicities [8]. Nowadays, more and more studies are focusing on the use of anthocyanins to prepare a pH indicator [2]. Anthocyanins are water soluble natural pigments in plants, and they are safe, nontoxic and sensitive to the change of pH value [9]. Vegetables and fruits can show a variety of colors, which is related to anthocyanins [10]. Because different kinds of vegetables and fruits have different pH values, anthocyanins can show different colors in different vegetables and fruits. Therefore, anthocyanins can be used as a natural pH-sensitive indicator to monitor pH changes [11]. Recently, some studies have reported anthocyanins from some vegetables and fruits such as *Phyllanthus reticulatus* [12], blueberry [13], black chokeberry [14] and the *Clitoria ternatea* flower [15] as the pH indicator to monitor the freshness of food. Anthocyanin extract can affect the structure and function of film [16]. At the same time, anthocyanins have a high degree of stability to ultraviolet light, which can increase the UV barrier properties of the film [17]. Coffee is an essential drink in daily life. The processing produces a large number of agricultural wastes such as coffee peels, which are rich in anthocyanins [18]. The *Robusta* Coffee is one of the most important coffee varieties in the world [19]. The research on the preparation of smart film with *Robusta* Coffee peel (RCP) as raw material has not been carried out. At the same time, the previous smart films have been studied a lot in monitoring meat deterioration, but there are few reports on monitoring the maturity of pickles. The maturity of pickles is crucial to the safety of consumers. Therefore, RCP extract can be used as an intelligent pH additive to prepare smart films for the pickle fermentation process and monitor the change in freshness of pickles.

In this study, CS–GL–RCP films was prepared by the solution casting method. By changing the content of RCP (0, 10%, 15% and 20%) in CS–GL films, the related performance indexes (color parameter, thickness, water content, swelling property, water vapor permeability, mechanical properties, water contact angle, UV-Vis light barrier property, microstructure and Fourier transform infrared (FTIR) analysis) of CS–GL–RCP films were studied. The optimal CS–GL–RCP indicator film was prepared to monitor the fermentation process of pickles within 15 days at 20 ± 1 °C, providing basic data for further research of smart films.

## 2. Material and Methods

### 2.1. Materials

*Robusta* Coffee and pickles were purchased from the market in Kunming, China. Chitosan, with a degree of deacetylation of 75%, was purchased from Jiazhi Biotechnology Co., Ltd. (Henan, China). Glycerol, with a density of 1.95 g/cm^3^, was purchased from Hengxing Chemical Reagent Co., Ltd. (Tianjin, China). All the chemicals have analytical purity grade.

### 2.2. Anthocyanins Extracted from RCP

The RCP extract was extracted according to a slight change in the literature [20]. The coffee peel was enucleated and dried in an oven for 48 h to remove moisture. Then, the peel was powdered and 1 g peel was added to 20 mL 75% ethanol. The solution sample underwent ultrasonic processing (80 W) for 30 min. Then, ethanol was removed with a rotary evaporator (RE-52CS; Shanghai Yarong Biochemical Instrument Factory, Shanghai, China) at 50 °C in the dark. The RCP solution was placed in the refrigerator at −20 °C for freezing, and the frozen solution was placed in the freeze dryer (Lab-1C-50E; Beijing Boyikang Experimental Instrument Co., Ltd., Beijing, China) for freezing drying (Degree of vacuum: 8.0 Pa) at −50 °C, so as to obtain the RCP powder. The RCP extract mixed with the pH solution of pH 1–14. The UV-Vis spectra were measured by the UV-Vis spectrophotometer (T9CS; Beijing Persee General Instrument Co., Ltd., Beijing, China).

### 2.3. UV-Vis Spectra of Anthocyanin Solution from RCP

The buffer solution with pH value of 1–14 was prepared by HCl solution and NaOH solution. One mL of buffer solution was mixed with 1 mL of RCP extract solution. The color changes of the solution were recorded by taking photos, and the UV-Vis spectrum of the solution was measured by the UV-Vis spectrophotometer (T9CS; Beijing Persee General Instrument Co., Ltd., Beijing, China). The measurement range was 450–800 nm [20].

### 2.4. Film Preparation

The chitosan–glycerol–*Robusta* Coffee peel (CS–GL–RCP) composite films were prepared using a solution casting method. The 2.0 g chitosan powder and 0.2 g glycerol were added to 100 mL acetic acid solution (2% *v*/*v*) and stirred for 12 h at 25 °C to obtain a chitosan solution (2% *w*/*v*) [5]. Next, the RCP extract was added to the mixed solution. The solution was poured into a polytetrafluoroethylene plate and dried in an oven at 35 °C, with a relative humidity of 50% for 12 h. When the content of RCP extract was 10%, 15% and 20%, based on the weight of chitosan, the prepared films were named CS–GL, and CS–GL–RCP10, CS–GL–RCP15 and CS–GL–RCP20, respectively.

### 2.5. Film Characterization

#### 2.5.1. Color

L, a (red/green) and b (yellow/blue) were tested by the colorimeter (WSC-S; Shanghai Precision Scientific Instrument Co., Ltd., Shanghai, China). Then, the chromatic aberration (Δ*E*) was calculated [21].
(1)$$\Delta E={\left[{\left(L*-L\right)}^{2}+{\left(a*-a\right)}^{2}+{\left(b*-b\right)}^{2}\right]}^{1/2}$$
where L* = 91.29, a* = −0.89 and b* = 3.76 represented values of a standard white plate.

#### 2.5.2. Thickness

The thickness was measured by a micrometer (DL9325; Shangqiu Shibang Trading Co., Ltd. Henan, China) with an accuracy of 0.01 mm. Five points were randomly selected on the membrane sample to calculate the average value.

#### 2.5.3. Water Content (WC)

The WC was measured by weight loss method [22]. The films (20 mm × 20 mm) were weighed as M_1_ and dried in the air dryer for 48 h at 75 °C. Then, the M_2_ of the dried film was weighed. The *WC* was calculated using the equations:(2)$$WC(\%)=100\times \frac{M_{1}-M_{2}}{M_{1}}$$

#### 2.5.4. Swelling Property (SP)

The films (20 mm × 20 mm) were dried for 48 h at 75 °C and weighed to obtain M_2_. The film was completely immersed in 50 mL deionized water for 24 h, and the water was removed and weighed to obtain M_3_ after drying for 48 h under the same conditions. The swelling ratio was calculated [23].
(3)$$swelling\;ratio\left(\%\right)=100\times \frac{M_{2}-M_{3}}{M_{2}}$$

#### 2.5.5. Water Vapor Permeability (WVP)

The WVP was tested by a weighing method according to the ASTM E96 standard and this paper [24]. A 5.0 g silica gel was put into the weighting test cup (relative humidity 0%). The cup was covered with the film and placed in a glass vacuum desiccator equipped with saturated sodium chloride solution (25 °C, relative humidity 95%) at the bottom. Due to the presence of saturated sodium chloride solution, the vapor pressure in the cup was considered to be zero. The external vapor pressure of the film was also obtained by the product of the relative humidity (95%) in the glass vacuum desiccator and the pure water vapor pressure at 25 °C. Therefore, a certain vapor pressure difference was maintained on both sides of the film. The sample was weighed every 12 h until the changes in weight were less than 0.001 g. The WVP was calculated from the changes in weight versus time, according to Equation (4).
(4)$$WVP=\frac{\Delta W\times l}{\Delta t\times A\times \Delta p}$$
where Δ*W* was the mass difference in time (g), l was the thickness of the film (m), Δ*t* was the water transmission time (s), A was the effective area of the film (m^2^), and Δ*p* was the water vapor pressure difference on both sides of the film.

#### 2.5.6. Mechanical Properties (MP)

The films (10 mm × 100 mm) were put into a rectangular spline. The tensile strength (TS) and elongation-at-break (EAB) were tested by the machine (SANS CMT 4104; MTS Systems Co., Ltd., Wuhan, China) according to GB/T1040-2006. The speed was 100 mm/min and initial distance between the two grippes was 80 mm. The measurement of each group of films was repeated at least five times [25].

#### 2.5.7. Water Contact Angle (WCA)

The films (20 mm × 20 mm) were flatly adhered to the glass sheet, and the glass sheet with the test sample was placed on the contact angle tester (JY-PHa; Chengde Jinhe Instrument Co., Ltd., Chengde, China) to test the WCA of the water on the sample. The 5 μL of water droplets were measured three times on each sample, and the average value was taken [26].

#### 2.5.8. UV-Vis Light Barrier Property

The films were sandwiched in the sample absorption tank of the UV-Vis spectrophotometer, so that the incident beam vertically passed the film. The measurement range was 200–800 nm [27].

#### 2.5.9. Microstructure

The cross sections were observed using a Scanning Electron Microscope (VEGA3; Tesken Co., Ltd., Prague, Czech Republic). The conductive gold layer of about 10 nm was plated on the fracture surface of the film [28].

#### 2.5.10. Fourier Transform Infrared (FTIR) Analysis

The samples were dried and fixed on the experimental platform for measuring film. The infrared absorption spectrum was tested by an ALPHA infrared spectrometer (Bruker, Germany). The setting test conditions were the spectral scanning range 4000–400 cm^−1^, resolution 4 cm^−1^, and scanning times 16 [29].

#### 2.5.11. Color Change at Different pH Values

The CS–GL–RCP15 films (20 mm × 20 mm) were immersed in 1 mL of solutions (pH = 1–14) for 5 min [30]. The color changes of the films were recorded by digital camera. 

### 2.6. Monitoring the Fermentation Process of Pickle

The monitoring of the fermentation process of pickle was modified according to the literature [31]. Fresh vegetables were selected and placed in a clean pot, salted and dried in the sun for 3 h until the vegetables had water exudation and the water was extruded. The vegetables were stored in a round pickle container (20 ± 1 °C). The boiling water was cooled and filled the container. The CS–GL–RCP15 film was placed on top as an indicator to monitor changes in pH values in the container. The film was photographed every five days. The color parameters were tested every five days, and Δ*E* was calculated by Equation (1). Then, the difference of Δ*E* was calculated as a color-difference value.

### 2.7. Statistical Analysis

The datas were analyzed using SPSS 22. The significant difference was determined using a One-Way ANOVA and Duncan analysis of variance with *p* < 0.05. 

## 3. Results and Discussion

### 3.1. Color Changes of RCP Extract

The color change of the RCP extract in the solution of pH 1–14 is shown in Figure 1. Coffee peel has been proven to be a source of anthocyanins [18]. Anthocyanins in the RCP extract were cyanidins, and the content of anthocyanins in the ethanol extract was 12.36 ± 0.31 mg/1 g dry coffee peel. The solutions with different pH values showed different color changes, mainly due to the transformation of the anthocyanin chemical structure [32]. The RCP extract is pink in a strong acidic medium due to the presence of yellow salt ions. In weak acidic medium, the color changes from pink to gray due to the formation of methanol pseudobase. In alkaline medium, yellow and green are the main colors, which may be due to the transformation of anthocyanin into a quinone-base structure in alkali [33]. The color of RCP extract can be significantly deepened in the neutral to acidic environment, indicating that the RCP extract was suitable as a pH indicator film coloring matter. The UV-Vis spectra of RCP extract in pH solutions is shown in Figure 2. At 525 nm, the RCP solution has the maximum absorption peak. The absorption peak moves to a higher wavelength due to the increase in pH values. The enhancement of color intensity and shift of the maximum absorption peak are related to the interaction between RCP and pH solutions [34]. Similar phenomena were found in anthocyanin-rich purple potato or roselle extracts [35].

### 3.2. Color of Film

The color parameters of film are shown in Table 1. The color of the CS–GL–RCP film is richer than CS–GL film by adding RCP extract. The a value and b value increased significantly (*p* < 0.05). The L value decreased significantly(*p* < 0.05) from 65.45 ± 0.01 to 59.66 ± 0.10, and the transparency also decreased. The Δ*E* value is very important in color change because it contains the relevant factors of color vision. With the addition of the RCP extract, the Δ*E* of the composite film increases significantly (*p* < 0.05). Mohammadalinejhad et al. found that when the difference of Δ*E* was greater than 5, the color difference was significant and could be observed by the naked eye [36]. The experimental results show that the CS–GL–RCP film is easy to be distinguished by the naked eye and can be used as a smart indicator packaging film. Comparing the three CS–GL–RCP composite films, the CS–GL–RCP15 film is the best choice for subsequent experiments.

### 3.3. Physicochemical Characterization

The physicochemical characterization of film was shown in Table 1. The thickness of the CS–GL film, CS–GL–RCP10 film, CS–GL–RCP15 film and CS–GL–RCP20 film are 38.0 ± 2.00, 41.6 ± 1.52, 47.2 ± 1.79 and 51.2 ± 1.92 μm, respectively. Anthocyanins contain hydroxyl groups, forming a complex structure with polymer molecules, thereby significantly increasing the thickness of the film (*p* < 0.05). The same results were obtained when CS film was mixed with mulberry extract [37].

It is worth noting that the water content (WC) of CS–GL films is the highest, and the WC of CS–GL–RCP films is slightly reduced compared to CS–GL film. With the increase in RCP extract, the WC of films declines significantly (*p* < 0.05) from 19.92% ± 0.55% to 11.09% ± 0.85%. The water content is an important index to evaluate the physical integrity of the film [34]. The anthocyanins in the RCP extract form hydrogen bonds with the hydrophilic groups in chitosan, which promotes the compatibility of anthocyanins with the film [38]. Anthocyanins limited the interaction between chitosan and water molecules, reduced the interaction between hydroxyl and amino groups, produced a dense and continuous structure of film, and formed the CS–GL–RCP film with lower WC [39]. Peralta et al. reported that the addition of aqueous hibiscus extract promoted a significant decrease in the WC of CS composite films (*p* < 0.05) [40].

The SP of the film is shown in Table 1. The CS–GL–RCP20 film has the lowest swelling ratio (185% ± 0.78%), and the CS–GL film has the highest swelling ratio (202% ± 1.28%). The SP of the CS–GL–RCP composite films become lower because of the interaction between RCP extract and chitosan molecules [41]. The swelling ratio represents the efficiency of the film in response to color. A high swelling ratio leads to the rapid release of anthocyanin solution, which is not conducive to the visual observation of the color change of smart indication packaging film with the changes of pH value [42]. Therefore, the CS–GL–RCP films with low swelling ratio are used for food packaging materials.

The function of food packaging is to reduce the water exchange between air and food. WVP represents the water vapor permeability of the film, so the lower the WVP value of the film, the better the waterproof performance of the film [43]. The WVP of composite films is shown in Table 1. After adding low content of RCP extract (10%), the WVP does not increase significantly (*p* > 0.05), but when the content is 15%, the WVP increases significantly (*p* < 0.05). Because the number of the RCP extract (10%) is limited, the impact on WVP of the composite film can be ignored. The anthocyanins in RCP extract contain a variety of phenolic hydroxyl hydrophilic substances, which change the original molecular interaction between CS and GL, and interfere with the dense structure of the film to a certain extent. At the same time, phenolic hydroxyl groups act as hydrophilic substances, making water easier to pass through the film [44]. The same change was reported in the article of black rice bran anthocyanins in chitosan-oxidized films [45].

### 3.4. Mechanical Properties of Film

The TS and EAB are shown in Table 1. With the increase in RCP extract, TS decreases significantly (*p* < 0.05) from 29.68 ± 0.29 to 11.36 ± 0.10 MPa, and EAB increases significantly (*p* < 0.05) from 9.28% ± 0.10% to 23.76% ± 0.21%. The addition of RCP extract changes the molecular interaction between CS and GL, promotes the movement of molecules, destroys the compact spatial structure of CS–GL film, and reduces the TS of the CS–GL–RCP films [45]. At the same time, the effect of RCP extract is similar to that of a plasticizer, which plays a plasticizing role and improves the EAB of the CS–GL–RCP films [46]. The same trends in mechanical performance were found in gum (ASKG) with *Artemisia sphaerocephala* Krasch extract [47].

### 3.5. Water Contact Angle of Film

The WCA of the film is tested by Angle Meter software and shown in Table 1. The WCA decreases significantly from 62.1 ± 1.84 to 41.9 ± 1.37 (*p* < 0.05) with the increase in RCP extract. The hydrophilicity of the film helps to intelligently indicate the color change of the smart film in response to pH changes in the environment [48]. Compared with CS–GL film, the WCA of CS–GL–RCP films decreases with the increase in RCP extract content because the anthocyanins in RCP extract have a large number of highly absorbent hydroxyl groups, making CS–GL–RCP films more hydrophilic [39]. The changes of WCA and WVP of the films are consistent, which can synergistically explain the related changes.

### 3.6. UV-Vis Light Barrier Property of Film

The visible light and ultraviolet light irradiate the food in the package, causing the food to deteriorate. Therefore, improving the UV resistance of the film has become a hot topic in the field of food packaging [49]. The UV-Vis light barrier property is shown in Figure 3. The UV transmittance of the composite films decreases significantly (*p* < 0.05) after adding the RCP extract. The transmittances of CS–GL–RCP15 and CS–GL–RCP20 films are close to 0 at the 200–350 nm. Because the CS–GL–RCP composite film contains the anthocyanins, anthocyanins have the potential to absorb ultraviolet light [48]. Ultraviolet rays increase the scattering reaction and refraction reaction when passing through the CS–GL–RCP composite film so that the ultraviolet transmittance of the film is greatly reduced, which is very important to protect the quality of food in the package [45]. Therefore, the CS–GL–RCP film can be used for food packaging because of its excellent light barrier.

### 3.7. Microstructure of Film

The cross section is shown in Figure 4. The CS–GL film showed the smoothest cross section and no wrinkles. With addition of RCP extract, the CS–GL–RCP films show typical images with porous structure [45]. The cross section of CS–GL–RCP films is affected by addition of RCP extract. This is due to the formation of new hydrogen bonds between the anthocyanin molecules and polymer chains [45]. The CS–GL–RCP films show a tight structure in the cross section diagram, indicating that the polymer and RCP extract had good compatibility and mixed solubility [41]. At the same time, the results confirm that RCP extract is evenly distributed on the films. In addition, the CS–GL–RCP20 film is rougher than other films and shows slight agglomeration. This may be because when the RCP extract reaches a certain high concentration, the solubility of the extract in the film-forming solution is limited. The results of the cross section are consistent with the physical properties.

### 3.8. FTIR Analysis of Film

As shown in Figure 5, the intermolecular interactions are examined by FTIR. The composite films have a wide band from 3395 to 3378 cm^−1^ (CH, alkyne stretching), 2961 to 2925 cm^−1^ (CH, alkane stretching), 1636 to 1606 cm^−1^ (C=C stretching), 1480 to 1460 cm^−1^ (C-H bend (in-plane), alkane), 1341 to 1300 cm^−1^ (C-H bend (in-plane), alkene), 1169 to 1129 cm^−1^ (C-C stretching), and 751 to 725 cm^−1^ (C-H bend (out of plane)). Because chitosan is a hydrophilic substance and the anthocyanins in RCP extract have a hydroxyl group, the wide band from 3395 to 3378 cm^−1^ (CH, alkyne stretching) of the CS–GL–RCP films observed is a hydrophilic substance. When the material is mixed, the physical mixing leads to the change of the spectral peak [50]. With the increase in the content of RCP extract in CS–GL–RCP films, the characteristic bands of CS–GL–RCP films have some displacement changes, which are due to the interaction between the hydroxyl group of the anthocyanin in the films and the amino group of the chitosan, which also confirms that the RCP extract is successfully fixed to the CS–GL film [51].

### 3.9. The Color Changes at Different pH Value of Film

The color change of the CS–GL–RCP15 film is shown in Figure 6. When the pH value increases from 1.0 to 14.0, the CS–GL–RCP15 film shows a visible color change from light red to green and yellow. Due to the structural changes of anthocyanins, the CS–GL–RCP15 film shows different color changes in solutions with different pH values [3]. The excessive anthocyanin content (20%) interferes with the color change of the composite film, so an appropriate amount of anthocyanin content (15%) is the best choice [3]. According to previous reports, many articles used the film with an anthocyanin extract content of 15% for preservation experiments [34,52].

### 3.10. Changes of the Fermentation Process of Pickle

The color change of CS–GL–RCP15 film during pickle storage for 0, 5, 10 and 15 days is shown in Figure 7. During storage, the color difference value of the CS–GL–RCP15 film changes as shown in Figure 8. Microbial activity is produced during the fermentation of pickles, and aerobic bacteria are inhibited in the anaerobic environment of pickles [53]. Therefore, the anaerobic bacteria *Lactobacillus* and anaerobic fungi of phylum *Ascomycota* found in fresh vegetables can be propagated and fermented [54]. *Lactobacillus* is considered to be the core bacteria in the fermentation process of pickles. At the same time, phylum *Ascomycota* plays a certain role in the fermentation process of pickles, and is considered to be the main genus of fungi in the fermentation process of pickles [55]. The fermentation maturity of pickles is closely related to the microflora [53]. The change of microflora in the environment leads to a change in pH values. Therefore, CS–GL–RCP15 film can monitor the fermentation process of pickles by color change. On day 0, the color of the CS–GL–RCP film was pink-white. At this time, the color of pickles was green, and the hardness of pickles was 42.69 ± 1.83. The pickles had the original astringent taste, so the quality could not reach the edible standard. With the increase in days, the color value of the CS–GL–RCP15 film significantly increased (*p* < 0.05). When Δ*E* of the composite films is higher than 5, the color change is easier to determine [36]. On day 10, the color change of the CS–GL–RCP15 film could be visually identified. On day 15, the color of the CS–GL–RCP15 film changed into pink-red, the color of the pickles was yellow. At the same time, the pickles had good flavor quality, strong fragrance, good texture and shape (hardness = 21.55 ± 0.98), and the quality of pickles reached the edible standard. The sensory evaluation steps of pickles are similar to those of Xiang et al. [56], and some sensory evaluations of pickles prove that the pickles are ripe. The CS–GL–RCP15 film is used to detect the change of pH in the pickle jars so as to monitor the fermentation process of pickles, so that people can judge whether pickles can be eaten through intuitive color change.

## 4. Conclusions

The CS–GL–RCP composite films were prepared by solvent evaporation. The results showed that RCP anthocyanins were successfully compounded onto CS–GL films and formed new interactions with CS–GL films. At different pH values, the color of the RCP extract is different and can be visually distinguished. With the addition of RCP extract, the physicochemical characterization and color parameters of the four films changed. The RCP extract increases the thickness and water vapor permeability (WVP) of the films. The RCP extract significantly reduces (*p* < 0.05) water content (WC), swelling property (SP) and water contact angle (WAC) of the films. Additionally, the CS–GL–RCP composite film has better flexibility and UV-Vis light barrier property than the CS–GL film. The CS–GL–RCP film is smart and its color changes significantly in the solution with different pH values. The color change of the film coincides with the maturation time point of the pickle. CS–GL–RCP15 film is recommended as a smart film to monitor the fermentation process of pickles stored at 20 ± 1 °C for 15 days, and it has broad application prospects in monitoring the maturity of the pickle during the fermentation process.

## Figures and Tables

**Figure 1 foods-12-02337-f001:**
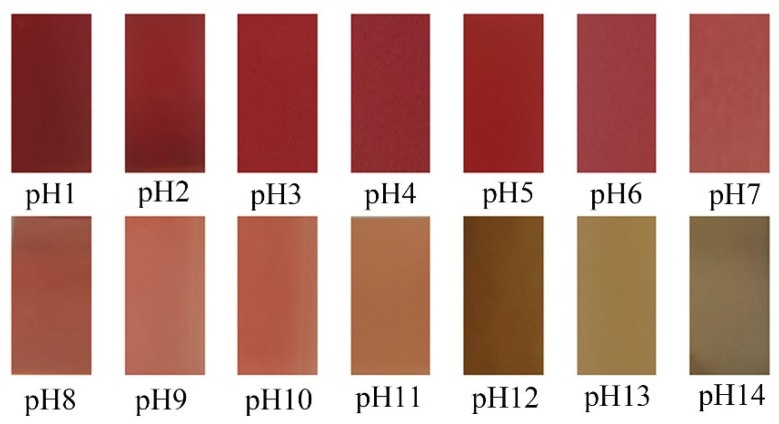
The color changes of RCP extract in different buffer solutions (pH = 1–14).

**Figure 2 foods-12-02337-f002:**
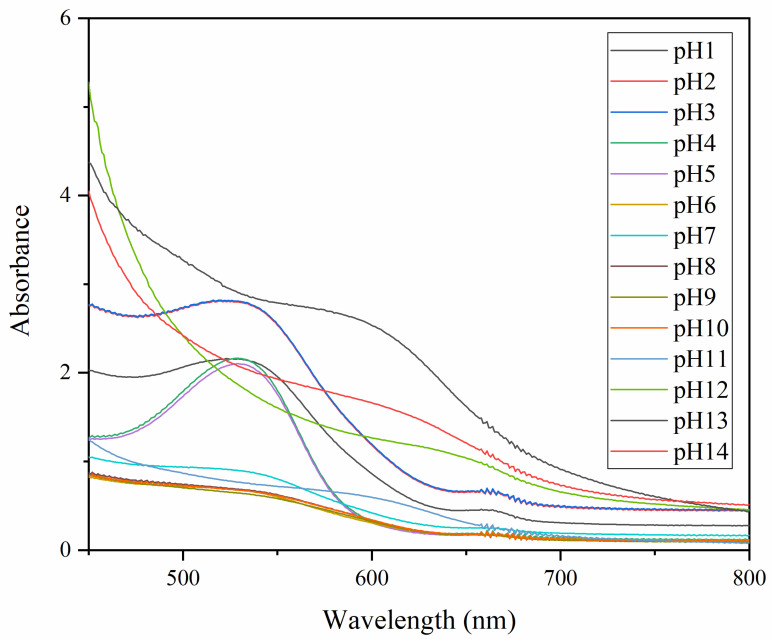
The UV-Vis spectra of RCP extract in different buffer solutions (pH = 1–14).

**Figure 3 foods-12-02337-f003:**
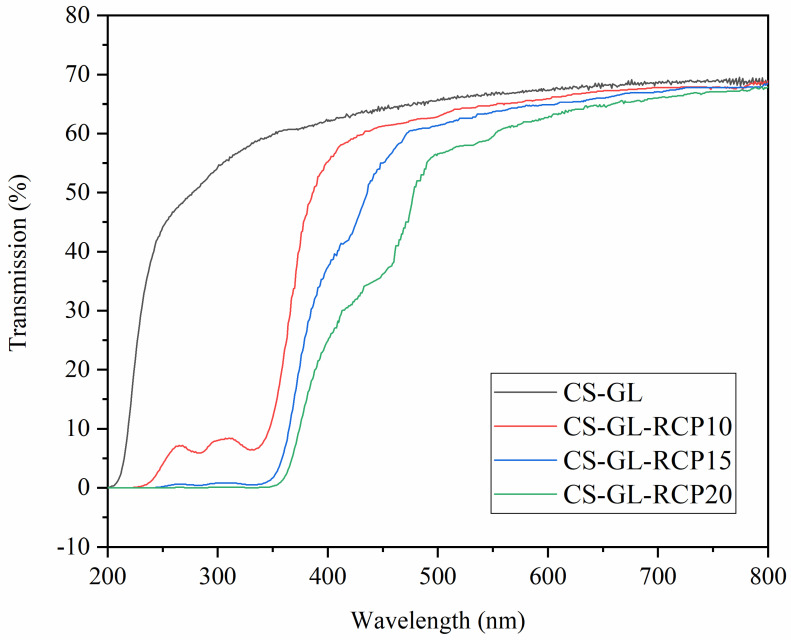
The UV-Vis light transmittance of CS−GL film and CS−GL−RCP composite films.

**Figure 4 foods-12-02337-f004:**
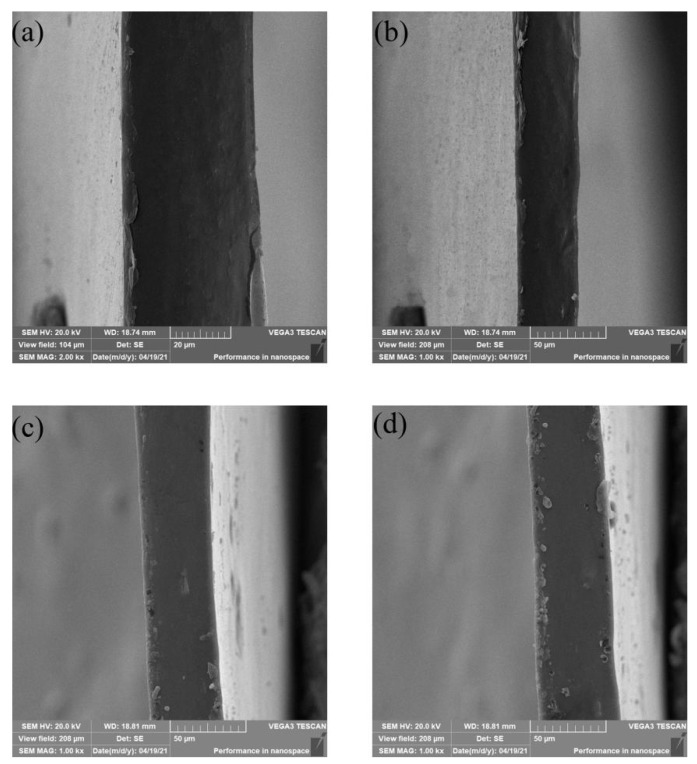
The SEM graphs of cross sections of (**a**) CS–GL film, (**b**) CS–GL–RCP10 film, (**c**) CS–GL–RCP15 film and (**d**) CS–GL–RCP20 film.

**Figure 5 foods-12-02337-f005:**
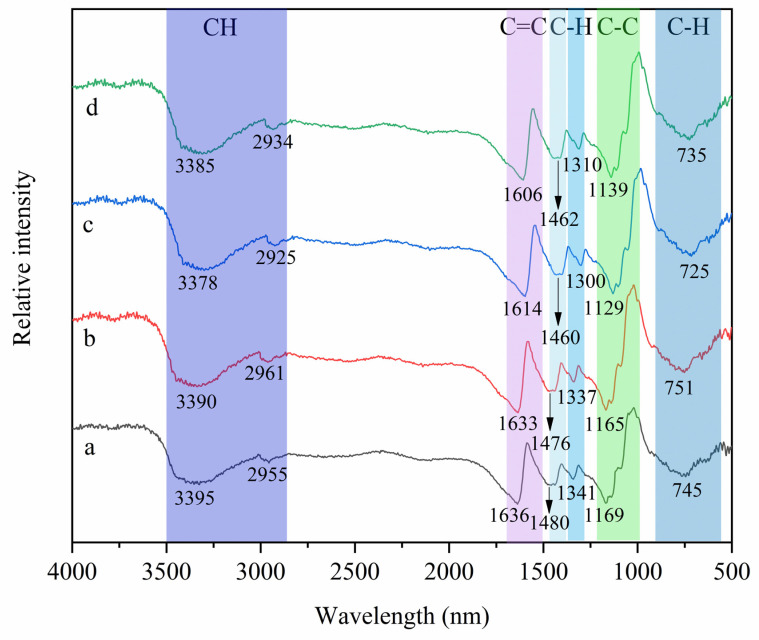
The FTIR spectra of (**a**) CS–GL film, (**b**) CS–GL–RCP10 film, (**c**) CS–GL–RCP15 film and (**d**) CS–GL–RCP20 film.

**Figure 6 foods-12-02337-f006:**
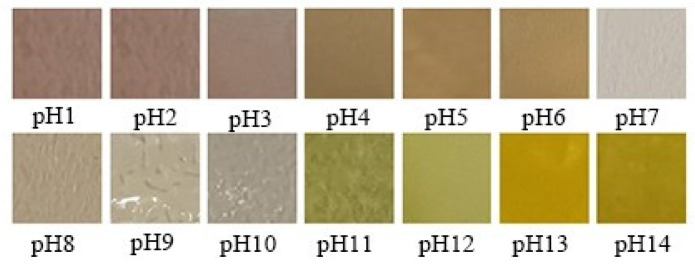
The color response of CS–GL–RCP15 film after immersing in buffer solution (pH = 1–14) for 5 min.

**Figure 7 foods-12-02337-f007:**
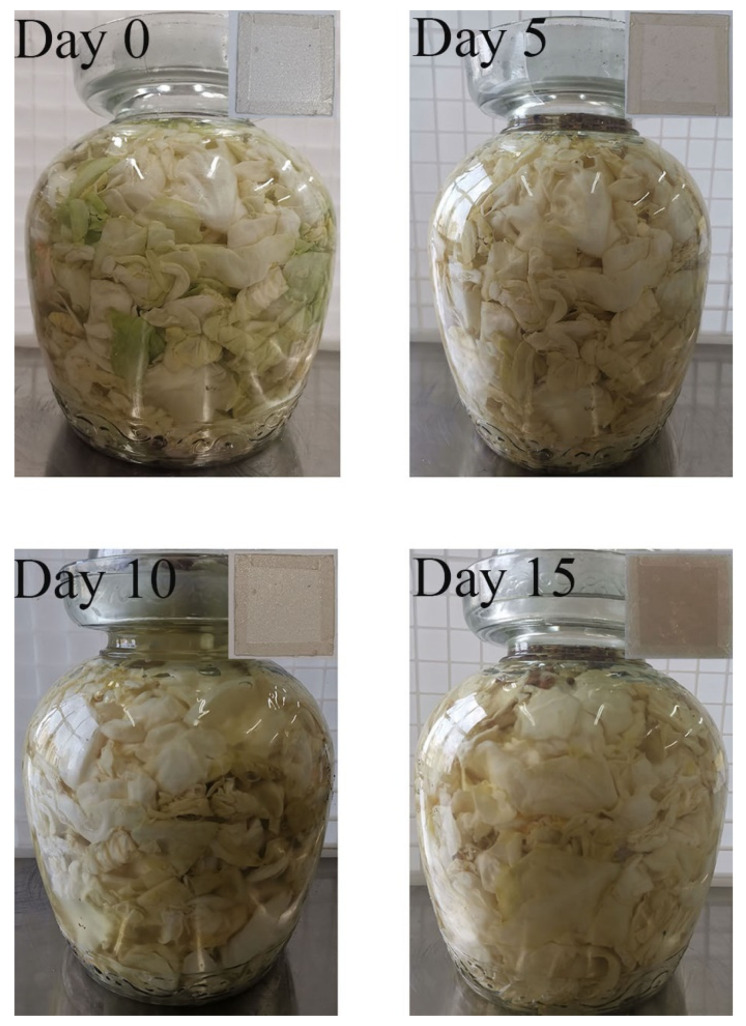
The color changes of CS–GL–RCP15 film on days 0, 5, 10 and 15.

**Figure 8 foods-12-02337-f008:**
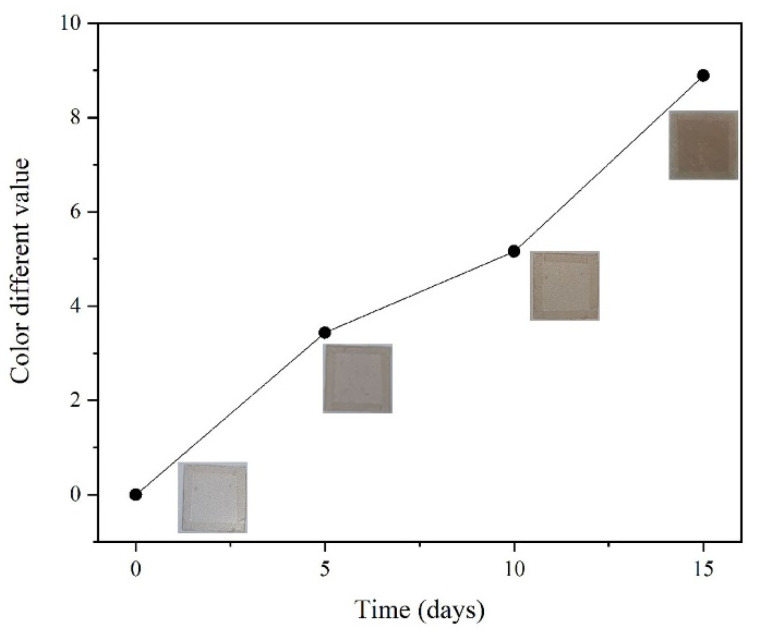
The color difference value of the CS–GL–RCP15 film on days 0, 5, 10 and 15.

**Table 1 foods-12-02337-t001:** The color parameters (L, a, b and Δ*E*), physical property (thicknesses, water content, swelling ratio, WVP and water contact angle) and mechanical property (tensile strength and elongation-at-break) of CS–GL films incorporated with RCP extract.

Film Sample	CS–GL	CS–GL–RCP10	CS–GL–RCP15	CS–GL–RCP20
L	65.45 ± 0.01 ^d^	62.91 ± 0.02 ^c^	61.88 ± 0.01 ^b^	59.66 ± 0.10 ^a^
a	3.38 ± 0.01 ^a^	4.21 ± 0.06 ^b^	6.74 ± 0.03 ^c^	8.93 ± 0.08 ^d^
b	4.42 ± 0.01 ^a^	7.86 ± 0.01 ^b^	9.91 ± 0.02 ^c^	10.56 ± 0.05 ^d^
Δ*E*	26.20 ± 0.01 ^a^	29.12 ± 0.01 ^b^	31.00 ± 0.01 ^c^	33.81 ± 0.08 ^d^
Thicknesses (μm)	38.0 ± 2.00 ^a^	41.6 ± 1.52 ^b^	47.2 ± 1.79 ^c^	51.2 ± 1.92 ^d^
Water content (%)	19.92 ± 0.55 ^d^	15.50 ± 0.23 ^c^	14.29 ± 0.07 ^b^	11.09 ± 0.85 ^a^
Swelling ratio (%)	202 ± 1.28 ^d^	195 ± 0.84 ^c^	191 ± 0.53 ^b^	185 ± 0.78 ^a^
WVP (10^−10^·g·m^−1^·s^−1^·Pa ^−1^)	1.87 ± 0.07 ^a^	1.92 ± 0.03 ^a^	2.11 ± 0.02 ^b^	2.29 ± 0.08 ^c^
Tensile strength (MPa)	29.68 ± 0.29 ^d^	21.56 ± 0.28 ^c^	16.69 ± 0.16 ^b^	11.36 ± 0.10 ^a^
Elongation at break (%)	9.28 ± 0.10 ^a^	13.72 ± 0.23 ^b^	18.68 ± 0.18 ^c^	23.76 ± 0.21 ^d^
Water contact angle (°)	62.1 ± 1.84 ^d^	56.9 ± 0.51 ^c^	48.3 ± 1.02 ^b^	41.9 ± 1.37 ^a^

The a–d values followed by different letters in the same column were significantly (*p* < 0.05) different, where a was the lowest value.

## Data Availability

The raw/processed data required to reproduce these findings cannot be shared at this time due to technical or time limitations.

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
