# Peer review of "Development of Smart Films of a Chitosan Base and *Robusta* Coffee Peel Extract for Monitoring the Fermentation Process of Pickles"

_foods, 2023, doi:10.3390/foods12122337_

Round 1
Reviewer 1 Report (New Reviewer)
The manuscript entitled: "Development of smart films based on chitosan-glycerol and Robusta Coffee peel extract for monitoring the fermentation process of pickle" is about preparation, characterization and application of an smart film based on chitosan and Robusta Coffee peel extract. In general, smart packaging is in trend currently and introducing different source of natural anthocyanins with evidence to use as smart packaging is interesting for researchers and industry. The research design is appropriate and the objectives of the manuscript well fit to the journal's aims and scopes.
There are some comments for authors to address before final decision by the editor as follows:
1- Title: Should be revise, no need to mention about glycerol in the title as it is common component of films.
2- Abstract: Rewrite the abstract, after objectives you need to mention about the treatments and methodology and after that bring the results. Shorten the results part and bring the major results, including some quantitative data.
3- Keywords: Good keywords, you can add "smart packaging"
4- Introduction: This part is OK, you need to extend the anthocyanin part and find recent publications/review about natural anthocyanins and their application in smart packaging. Also this article is related to this field and you can use in both introduction and discussion part: Hematian, Fahimeh, et al. "Preparation and characterization of an intelligent film based on fish gelatin and Coleus scutellarioides anthocyanin to monitor the freshness of rainbow trout fish fillet." Food Science & Nutrition 11.1 (2023): 379-389.
5- Materials: This part is OK, all about source of other chemicals used in the research or mention all analytical grades.
6- Methods: All methods should have a proper reference(s) from standard or published article.
7- Results and discussion: This part is OK, something unclear is about anthocyanin content of the extract. You need to measure the anthocyanin content and add the results to make it comparable with similar works in this area.
8- Conclusions:
Rewrite this part, no need to start by "In general". Justify research hypothesis and if there is any research recommendation.
Author Response
Response to reviewer’s comments
Manuscript ID: Foods-2341145
Title: Development of smart films based on chitosan-glycerol and Robusta Coffee peel extract for monitoring the fermentation process of pickle.
Journal: Foods.
Dear reviewer,
I have made revisions according to reviewer's comments. It had been appended below. The revised section was shown in red color.
Yours sincerely,
Yuyue Qin
Reviewer comments:
Reviewer: The manuscript entitled: "Development of smart films based on chitosan-glycerol and Robusta Coffee peel extract for monitoring the fermentation process of pickle" is about preparation, characterization and application of an smart film based on chitosan and Robusta Coffee peel extract. In general, smart packaging is in trend currently and introducing different source of natural anthocyanins with evidence to use as smart packaging is interesting for researchers and industry. The research design is appropriate and the objectives of the manuscript well fit to the journal's aims and scopes.
Revision: We have revised the manuscript according to your comments. Thank you very much!
Specific comment:
Point 1: 1-Title: Should be revise, no need to mention about glycerol in the title as it is common component of films.
Revision: Page 1, line 2-4: The title“Development of smart films of chitosan-based and Robusta Coffee peel extract for monitoring the fermentation process of a pickle”has been modified. Thank you very much!
Point 2: 2-Abstract: Rewrite the abstract, after objectives you need to mention about the treatments and methodology and after that bring the results. Shorten the results part and bring the major results, including some quantitative data.
Revision: Page 1, line 13-27: By changing the content of RCP (0, 10%, 15% and 20%) in CS-GL film, the related performance indicators of CS-GL-RCP films were studied. The results showed that the CS-GL-RCP films had excellent mechanical properties, and CS-GL-RCP-15 film maintained the excellent tensile strength (TS) of 16.69 MPa and elongation at break (EAB) of 18.68% with RCP extract. CS-GL-RCP films had the best UV-vis light barrier property at 200-350 nm and the UV transmittance was close to 0. The microstructure observation results showed that CS-GL-RCP films had a dense and uniform cross section, which proved that RCP extract had good compatibility with the polymer. In addition, CS-GL-RCP15 film was pH-sensitive and could exhibit different color changes with different pH solutions. So, the CS-GL-RCP15 film was used to detect the fermentation process of pickles at 20±1 °C for 15 days. The pickles were stored in a round pickle container after boiling water cooled. The color of CS-GL-RCP15 film changed significantly, which was consistent with the change of pickles from fresh to mature. The color of the smart film changed significantly with the maturity of pickles and the difference of ΔE of film increased to 8.89 (15 Day), which can be seen by the naked eye. Therefore, CS-GL-RCP films prepared in this study provided a new strategy for the development of smart packaging materials. The results part has been shorted and the major results have been brought. The treatments and methodology of pickles and some quantitative data have been added. Thank you very much!
Point 3: 3-Keywords: Good keywords, you can add "smart packaging"
Revision: Page 1: The keyword "Smart packaging" has been added. Thank you very much!
Point 4: 4-Introduction: This part is OK, you need to extend the anthocyanin part and find recent publications/review about natural anthocyanins and their application in smart packaging. Also this article is related to this field and you can use in both introduction and discussion part: Hematian, Fahimeh, et al. "Preparation and characterization of an intelligent film based on fish gelatin and Coleus scutellarioides anthocyanin to monitor the freshness of rainbow trout fish fillet." Food Science & Nutrition 11.1 (2023): 379-389.
Revision: Page 2, line 49-73: With the progress of science and the development of the times, the safety of food additives has attracted more and more attention [7]. Because natural additives are safer and more environmentally friendly than chemical synthetic dyes. The development and utilization of natural additives have become the general trend [8]. Nowadays, more and more studies were focusing on the use of anthocyanins to prepare a pH indicator [2]. Anthocyanins are water soluble natural pigment in plants and they are safe, nontoxic and sensitive to the change of pH value [9]. Vegetables and fruits can show a variety of colors, which is related to anthocyanins [10]. Because different kinds of vegetables and fruits have different pH values, anthocyanins can show different colors in different vegetables and fruits. Therefore, anthocyanins can be used as a natural pH sensitive indicator to monitor pH changes [11]. Recently, some studies have reported the anthocyanins from some vegetables and fruits such as Phyllanthus reticulatus [12], blueberry [13], black chokeberry [14], and clitoria ternatea flower [15] as the color changes of the films frequently. Anthocyanin extract can affect the structure and function of film [16]. At the same time, anthocyanins have a high degree of stability to ultraviolet light, which can increase the UV barrier properties of the film [17]. Coffee is an essential drink in daily life. The processing produces a large number of agricultural wastes such as coffee peel, which is rich in anthocyanins [18]. The Robusta Coffee is one of the most important coffee varieties in the world [19]. The research on the preparation of smart film with Robusta Coffee peel (RCP) as raw material has not been carried out. At the same time, the previous smart films have been studied a lot in monitoring meat deterioration, but there are few reports on monitoring the maturity of pickles. The maturity of pickles is crucial to the safety of consumers. Therefore, RCP extract can be used as an intelligent pH additive to prepare smart films for pickle fermentation process and monitor the freshness change of pickles. The anthocyanin part has been extended and this article has been added. Thank you very much!
- Hematian,F.;Baghaei, H.; Nafchi, A.M.; Bolandi, M. Preparation and characterization of an intelligent film based on fish gelatin and Coleus scutellarioides anthocyanin to monitor the freshness of rainbow trout fish fillet. Food Sci Nutr. 2023, 379-389.
Point 5: 5-Materials: This part is OK, all about source of other chemicals used in the research or mention all analytical grades.
Revision: Page 2, line 85-89: Robusta Coffee and pickle were purchased from the market in Kunming, China. Chitosan with the degree of deacetylation 75% was purchased from Jiazhi Biotechnology Co., Ltd. (Henan, China). Glycerol with the density of 1.95 g/cm3 was purchased from Hengxing Chemical Reagent Co., Ltd. (Tianjin, China). All the chemicals have analytical purity grade. We have revised the materials according to your comments. Thank you very much!
Point 6: 6-Methods: All methods should have a proper reference(s) from standard or published article.
Revision: Page 4, line 171-172: The conductive gold layer of about 10 nm was plated on the fracture surface of the film [28]. Page 4, line 176-177: The setting test conditions were spectral scanning range 4000-400 cm-1, resolution 4 cm-1, and scanning times 16 [29]. Page 4, line 179-180: The CS-GL-RCP15 films (20 mm × 20 mm) were immersed in 1 mL of solutions (pH=1-14) for 5 min [30]. Page 5, line 182-183: The monitoring the fermentation process of pickle was modified according to the literature [31]. All methods have been added a proper reference(s) from standard or published article. Thank you very much!
- Hu,D.;Liu, X.; Qin, Y.; Yan, J.; Yang, Q. A novel intelligent film with high stability based on chitosan/sodium alginate and coffee peel anthocyanin for monitoring minced beef freshness. Int J Food Sci Tech. 2022, 57.
- Chen,K.;Li, J.; Li, L.; Wang, Y.; Qin, Y.; and Chen, H. A pH indicator film based on sodium alginate/gelatin and plum peel extract for monitoring the freshness of chicken. Food Biosci. 2023, 102584.
- Zhu,B.; Lu,W.; Qin, Y.; Cheng, G.; Yuan, M.; Li, L. An intelligent pH indicator film based on cassava starch/polyvinyl alcohol incorporating anthocyanin extracts for monitoring pork freshness. J Food Process Pres. 2021, e15822.
- Hu,D.;Liu, X.; Qin, Y.; Yan, J.; Li, J.; Yang, Q. A novel edible packaging film based on chitosan incorporated with persimmon peel extract for the postharvest preservation of banana. Food Qual Saf. 2022.
Point 7: 7-Results and discussion: This part is OK, something unclear is about anthocyanin content of the extract. You need to measure the anthocyanin content and add the results to make it comparable with similar works in this area.
Revision: Page 5, line 195-197: Coffee peel has been proven to be a source of anthocyanin [18]. Anthocyanins in RCP extract were cyanidins, and the content of anthocyanins in ethanol extract was 12.36±0.31 mg/1 g dry coffee peel. We have revised the materials according to your comments. Thank you very much!
Point 8: 8-Conclusions: Rewrite this part, no need to start by "In general". Justify research hypothesis and if there is any research recommendation.
Revision: Page 13-14, line 377-390: The CS-GL-RCP composite films were prepared by solvent evaporation. The results showed that RCP anthocyanins were successfully compounded onto CS-GL films and formed new interactions with CS-GL films. At different pH values, the color of the RCP extract is different and can be visually distinguished. With the addition of RCP extract, the physicochemical characterization and color parameters of the four films changed. The RCP extract increases the thickness and water vapor permeability (WVP) of the films. The RCP extract obviously reduces (p<0.05) water content (WC), swelling property (SP) and water contact angle (WAC) of the films. In addition, the CS-GL-RCP composite film has better flexibility and UV-Vis light barrier property than the CS-GL film. The CS-GL-RCP film is smart and its color changes obviously in the solution with different pH values. The color change of the film is in good agreement with the maturation time point of the pickle. CS-GL-RCP15 film is recommended as a smart film to monitor the fermentation process of pickles stored at 20±1 °C for 15 days and has broad application prospects in monitoring the maturity of pickle fermentation process. We have revised the conclusions according to your comments. Thank you very much!

Reviewer 2 Report (New Reviewer)
This paper dealing with the development of new smart films based on chitosan and anthocyanin-rich Robusta coffee peel extract. This paper gives an original data dealing with impact of RCP for monitoring the fermentation process of pickle. But unfortunately the relationships between the reported properties of the extract and the interactions with chitosan must be more explained. somethings need to be more explained and improved (details in the following questions and remarks). The effect of water content is tremendously known on structural properties of hydrocolloid based films, but the moisture content of the different film recipes was not given. Moreover, even if English is not my mother language, the manuscript needs English grammar and spelling improvement. A lot of contraction in this manuscript, very week explanation of the results. The manuscript is a series of analyses without any link between different parts. I very week discussion of results with the literature. A leak of explanation of plasticizer effect of RCP on films matrix, and it’s relation to structural and microstructural and barrier properties. So I recommend more explanation and a lot of reflexing point and correction to this paper that need to be improved.
Abstract:
How you can explain the increasing of WVP by RCP adding and the reducing of swelling properties. In the case of hydrocolloid-based film if film swell, the absorb water and therefore the chain mobility increase and the WVP increase.
Line 24:
“Therefore, the CS-GL-RCP film prepared in this study provided a new strategy for the development of active packaging materials”. For me the film is more smart than active. Did you check the bioactivity of films?
Introduction:
Line 62-64: please re-write this sentence: the objective should be clear. Addition of RCP to CS films in order to evaluate the mechanical (TS, EAB), barrier and water content as well as swelling properties and then microstructure properties.
Material and methods:
Line 72: please add the glycerol density.
Line 84: “The UV-Vis spectra measured by the UV-Vis spectrophotometer (T9CS, Beijing Persee General Instrument Co., Ltd. Beijing, China)”: remove this sentence from this part concerning the anthocyanin extraction. Insert it the part of material characterization (UV test).
Line 92: the % of RCP (0,10, 15 and 20% is w/w? wt of polymer without glycerol or with glycerol? Or w/v?? Please be more precise?
Line 114: “5.0 g anhydrous calcium chloride was put into the weighting test cup”: in order to fixe RH at????.
L115-117: “The cup was covered with the film and placed in the dryer equipped with saturated sodium chloride solution (25°C, relative humidity 75%) and certain vapor pressure difference was maintained on both sides of the film”: please be more precise: firstly, dryer equipped with ….: is climatic room??? What is the dryer used? The in order to maintained certain vapor pressure difference on both sides of the film; what id the ΔRH gradient: 75-X%?? Please precise.
Line 127: Is sample were stored at RH condition before the mechanical test? This later is highly dependent on RH condition, so sample should be equipped on RH condition before mechanical testing, in general one week at 50% RH. The initial distance between the two grippes?
Line 143: is the FTIR analysis was done by ATR (Attenuated Total Reflectance accessory)?
Line 158: The dates were analyzed using SPSS 22. The significant difference was used Duncan analysis of variance with p<0.05: is the test is T-test? Or one-way annova?.
Result and discussion:
Line 165: The changes in color of RCP extract are caused by the structural change of anthocyanins: please explain.
Line 183: The L value is decreased obviously (p<0.05) from 65.45 to 59.66: please add SD.
Line 198: 3.3. Thickness, WC, SP, and WVP of film: you can put as a title of 3.3: physicochemical characterization.
Line 205-209: With the increased of anthocyanins content, the WC of film declines obviously (p<0.05) from 19.92 to 11.09. The main reason for the formation of water content is the molecular interaction between water molecules and polymer [34]. The hydrogen bonds are generated in the RCP extract and the hydrophilic groups in the chitosan, so that the interaction of the chitosan with the water molecules is weakened to form a denser film with a lower water content [35]. I’m not sure for this explanation: id we have interaction between molecules and CS, this later become more related to water and film become less dense and WC increase.
Line 225-227: Anthocyanin in RCP extract contains multiple phenolic hydroxyl groups as hydrophilic substances, making it easier for water to pass through the films [39]. The same change was reported in the article of black rice bran anthocyanins in chitosan/oxidized films [40]. Yes but you forget the OH group of anthocyanins can make some electrostatic interaction with NH3+ of chitosan and therefore denser the films network and limit water transfer through the film matrix. Please explain more the mechanisms. The PH of film forming solution?
Line 232-233: The combination of RCP molecules and chitosan blocks the interaction between chitosan and glycerol, which reduces the TS of the composite film: NO, if we block the interaction between chitosan and glycerol the EAB is more affected. Be sure that the plasticizer affects more the EAB and YM and the addition of crosslinker affect more the TS. In addition, this sentence is in contradiction with the line 236: The effect of RCP extract is similar to that of a plasticizer and improves mechanical properties of the film: if mechanical properties is improved TS is improved. In your sentence you can say that RCP play a plasticizer role so the EAB increased. Pease add the SD to all value presented in the discussion part.
Line 244: Due to the presence of some hydroxyl groups in the anthocyanin in the RCP extract, the WCA decreases after adding the RCP extract [35]. Please explain more. In your case you should do the Critical surface tension and free tension energy with polar and dispersive compounds used 3 or 4 liquids.
Line 274 the FTIR part should be revised: If we have some interaction between OH group of anthocyanin and NH3+ of chitosan you van get some shift relative to CH region.
In addition, in the figure of FTIR spectra please add the involved group of each peak. One example NH-amide II band 1540-1570; CO amide I in 1630-1660…..
Conclusion
L323: should be more consistent and different to abstract. Add also the perspective.
Author Response
Response to reviewer’s comments
Manuscript ID: Foods-2341145
Title: Development of smart films based on chitosan-glycerol and Robusta Coffee peel extract for monitoring the fermentation process of pickle.
Journal: Foods.
Dear reviewer,
I have made revisions according to reviewer's comments. It had been appended below. The revised section was shown in red color.
Yours sincerely,
Yuyue Qin
Reviewer comments:
Reviewer: This paper dealing with the development of new smart films based on chitosan and anthocyanin-rich Robusta coffee peel extract. This paper gives an original data dealing with impact of RCP for monitoring the fermentation process of pickle. But unfortunately the relationships between the reported properties of the extract and the interactions with chitosan must be more explained. somethings need to be more explained and improved (details in the following questions and remarks). The effect of water content is tremendously known on structural properties of hydrocolloid based films, but the moisture content of the different film recipes was not given. Moreover, even if English is not my mother language, the manuscript needs English grammar and spelling improvement. A lot of contraction in this manuscript, very week explanation of the results. The manuscript is a series of analyses without any link between different parts. I very week discussion of results with the literature. A leak of explanation of plasticizer effect of RCP on films matrix, and it’s relation to structural and microstructural and barrier properties. So I recommend more explanation and a lot of reflexing point and correction to this paper that need to be improved.
Revision: We have revised the manuscript according to your comments. Thank you very much!
Specific comment:
Point 1: Abstract:
How you can explain the increasing of WVP by RCP adding and the reducing of swelling properties. In the case of hydrocolloid-based film if film swell, the absorb water and therefore the chain mobility increase and the WVP increase.
Revision: Page 1, line 13-14: By changing the content of RCP (0, 10%, 15% and 20%) in CS-GL film, the related performance indicators of CS-GL-RCP films were studied. RCP can increase the interaction between chitosan molecules and inhibit water diffusion into dry films to reduce swelling properties [36]. Anthocyanin in RCP extract contains multiple phenolic hydroxyl groups as hydrophilic substances, making it easier for water to pass through the films [39], thus increasing the WVP value. The major results have been brought and other results part has been shorted. Thank you very much!
Point 2: Line 24:
“Therefore, the CS-GL-RCP film prepared in this study provided a new strategy for the development of active packaging materials”. For me the film is more smart than active. Did you check the bioactivity of films?
Revision: Page 1, line 26-27: Therefore, CS-GL-RCP films prepared in this study provided a new strategy for the development of smart packaging materials. The “smart packaging materials”has been modified. Because of the use of biodegradable chitosan as film matrix, this paper did not check the bioactivity of films, and next papers have been checked the bioactivity of films. Thank you very much!
Point 3: Introduction:
Line 62-64: please re-write this sentence: the objective should be clear. Addition of RCP to CS films in order to evaluate the mechanical (TS, EAB), barrier and water content as well as swelling properties and then microstructure properties.
Revision: Page 2, line 74-82: In this study, CS-GL-RCP films was prepared by solution casting method. By changing the content of RCP (0, 10%, 15% and 20%) in CS-GL films, the related performance indexes (color parameter, thickness, water content, swelling property, water vapor permeability, mechanical properties, water contact angle, UV-Vis light barrier property, microstructure and Fourier transform infrared (FTIR) analysis) of CS-GL-RCP films were studied. At different pH values, the CS-GL-RCP films have different color changes. The excellent CS-GL-RCP15 film was prepared to monitor the fermentation process of pickles within 15 days at 20±1 °C, providing basic data for further research of smart films. We have revised the introduction according to your comments. Thank you very much!
Point 4: Material and methods:
Line 72: please add the glycerol density.
Revision: Page 2, line 87-88: Glycerol with the density of 1.95 g/cm3 was purchased from Hengxing Chemical Reagent Co., Ltd. (Tianjin, China). We have revised the material according to your comments. Thank you very much!
Point 5: Line 84:“The UV-Vis spectra measured by the UV-Vis spectrophotometer (T9CS, Beijing Persee General Instrument Co., Ltd. Beijing, China)”: remove this sentence from this part concerning the anthocyanin extraction. Insert it the part of material characterization (UV test).
Revision: Page 3, line 103-108: 2.3 UV-Vis spectra of anthocyanin solution from RCP
The buffer solution with pH value of 1-14 was prepared by HCl solution and NaOH solution. 1 mL of buffer solution was mixed with 1 mL of RCP extract solution. The color changes of the solution were recorded by taking photos, and the UV-Vis spectrum of the solution was measured by the UV-Vis spectrophotometer (T9CS, Beijing Persee General Instrument Co., Ltd. Beijing, China). The measurement range was 450-800 nm [20]. We have revised the methods according to your comments. The UV-Vis spectra of the RCP extract was re-listed for a chapter, because the subsequent UV test was to test the UV transmittance of the film. Thank you very much!
Point 6: Line 92: the % of RCP (0,10, 15 and 20% is w/w? wt of polymer without glycerol or with glycerol? Or w/v?? Please be more precise?
Revision: Page 3, line 116-118: When the content of RCP extract was 10%, 15% and 20% based on the weight of chitosan, the prepared films were named CS-GL, CS-GL-RCP10, CS-GL-RCP15 and CS-GL-RCP20, respectively. We have revised the methods according to your comments. Thank you very much!
Point 7: Line 114: “5.0 g anhydrous calcium chloride was put into the weighting test cup”: in order to fixe RH at????.
Revision: Page 4, line 139-143: The WVP tested by weighing method according ASTM E96 standard and this paper [24]. 5.0 g anhydrous calcium chloride was put into the weighting test cup (relative humidity 0%). The cup was covered with the film and placed in a glass vacuum desiccator equipped with saturated sodium chloride solution (25 °C, relative humidity 95%) at the bottom.“5.0 g anhydrous calcium chloride was put into the weighting test cup”: in order to calculate the WVP of the film by increasing the weight of the film by absorbing water. The relative humidity of weighting test cup is 0% . We have revised the methods according to your comments. Thank you very much!
Point 8: L115-117: “The cup was covered with the film and placed in the dryer equipped with saturated sodium chloride solution (25°C, relative humidity 75%) and certain vapor pressure difference was maintained on both sides of the film”: please be more precise: firstly, dryer equipped with ….: is climatic room??? What is the dryer used? The in order to maintained certain vapor pressure difference on both sides of the film; what id the ΔRH gradient: 75-X%?? Please precise.
Revision: Page 4, line 139-147: The WVP tested by weighing method according ASTM E96 standard and this paper [24]. 5.0 g anhydrous calcium chloride was put into the weighting test cup (relative humidity 0%). The cup was covered with the film and placed in a glass vacuum desiccator equipped with saturated sodium chloride solution (25 °C, relative humidity 95%) at the bottom. Due to the presence of saturated sodium chloride solution, the vapor pressure in the cup is considered to be zero. The external vapor pressure of the film is also obtained by the product of the relative humidity (95%) in the glass vacuum desiccator and the pure water vapor pressure at 25 °C. Therefore, a certain vapor pressure difference is maintained on both sides of the film. Dryer is a glass vacuum desiccator. The relative humidity of weighting test cup is 0%. The cup was covered with the film and placed in a glass vacuum desiccator equipped with saturated sodium chloride solution (25 °C, relative humidity 95%) at the bottom. Due to the presence of saturated sodium chloride solution, the vapor pressure in the cup is considered to be zero. The external vapor pressure of the film is also obtained by the product of the relative humidity (95%) in the glass vacuum desiccator and the pure water vapor pressure at 25 °C. Therefore, a certain vapor pressure difference is maintained on both sides of the film. We have revised the methods according to your comments. Thank you very much!
Point 9: Line 127: Is sample were stored at RH condition before the mechanical test? This later is highly dependent on RH condition, so sample should be equipped on RH condition before mechanical testing, in general one week at 50% RH. The initial distance between the two grippes?
Revision: Page 4, line 156-157: The speed was 100 mm/min and initial distance between the two grippes was 80 mm. As you said, RH has a great influence on the mechanical properties of the film, so we immediately tested the mechanical properties of the film after preparation. We have revised the methods according to your comments. Thank you very much!
Point 10: Line 143: is the FTIR analysis was done by ATR (Attenuated Total Reflectance accessory)?
Revision: Page 4, line 174-177: The samples were dried and fixed on the experimental platform for measuring film. The infrared absorption spectrum was tested by an ALPHA infrared spectrometer (Bruker, Germany). The setting test conditions were spectral scanning range 4000-400 cm−1, resolution 4 cm-1, and scanning times 16 [29]. The FTIR analysis was not done by ATR (Attenuated Total Reflectance accessory). Thank you very much!
- Chen,K.;Li, J.; Li, L.; Wang, Y.; Qin, Y.; and Chen, H. A pH indicator film based on sodium alginate/gelatin and plum peel extract for monitoring the freshness of chicken. Food Biosci. 2023, 102584.
Point 11: Line 158: The dates were analyzed using SPSS 22. The significant difference was used Duncan analysis of variance with p<0.05: is the test is T-test? Or one-way annova?.
Revision: Page 5, line 190-191: The significant difference was used One Way ANOVA and Duncan analysis of variance with p<0.05.
Point 12: Result and discussion:
Line 165: The changes in color of RCP extract are caused by the structural change of anthocyanins: please explain.
Revision: Page 5, line 198-203: RCP extract is pink in a strong acidic medium due to the presence of yellow cations. In weak acidic medium, the color changes from pink to gray due to the formation of methanol pseudobase and acidic hydroxyl. In alkaline medium, yellow and green are the main colors, which may be due to the transformation of anthocyanin into quinone base structure in weak alkali. The anthocyanins change from flavonoid cation to methanol base and acidic hydroxyl [33]. We have revised the result according to your comments. Thank you very much!
Point 13: Line 183: The L value is decreased obviously (p<0.05) from 65.45 to 59.66: please add SD.
Revision: Page 6, line 219-220: The L value is decreased obviously (p<0.05) from 65.45±0.01 to 59.66±0.10, the transparency is decreased. We have revised the result according to your comments. Thank you very much!
Point 14: Line 198: 3.3. Thickness, WC, SP, and WVP of film: you can put as a title of 3.3: physicochemical characterization.
Revision: Page 7, line 235-236: 3.3. Physicochemical characterization
The physicochemical characterization of film was shown in Table 1.We have revised the title according to your comments. Thank you very much!
Point 15: Line 205-209: With the increased of anthocyanins content, the WC of film declines obviously (p<0.05) from 19.92 to 11.09. The main reason for the formation of water content is the molecular interaction between water molecules and polymer [34]. The hydrogen bonds are generated in the RCP extract and the hydrophilic groups in the chitosan, so that the interaction of the chitosan with the water molecules is weakened to form a denser film with a lower water content [35]. I’m not sure for this explanation: id we have interaction between molecules and CS, this later become more related to water and film become less dense and WC increase.
Revision: Page 7, line 247-252: Anthocyanins in RCP extract produce hydrogen bonds with the hydrophilic groups in chitosan. Anthocyanins limit the interaction between chitosan and water molecules and reduce the interaction between hydroxyl and amino groups, so the CS-GL-RCP film with lower WC is formed [39]. Peralta et al. reported that the addition of Aqueous hibiscus extract promoted a significant decrease in the WC of CS composite films (p<0.05) [40]. We have revised the result according to your comments and added reference with the same WC results of CS films. Thank you very much!
- Peralta,J.;Bitencourt-Cervi, C.M.; Maciel, V.B.V.; Yoshida, C.M.P.; Carvalho, R.A. Aqueous hibiscus extract as a potential natural pH indicator incorporated in natural polymeric films. Food Packaging Shelf. 2019, 47-55.
Point 16: Line 225-227: Anthocyanin in RCP extract contains multiple phenolic hydroxyl groups as hydrophilic substances, making it easier for water to pass through the films [39]. The same change was reported in the article of black rice bran anthocyanins in chitosan/oxidized films [40]. Yes but you forget the OH group of anthocyanins can make some electrostatic interaction with NH3+ of chitosan and therefore denser the films network and limit water transfer through the film matrix. Please explain more the mechanisms. The PH of film forming solution?
Revision: Page 7, line 268-272: The anthocyanins in RCP extract contain a variety of phenolic hydroxyl hydrophilic substances, which change the original molecular interaction between CS and GL, and interfere with the dense structure of the film to a certain extent. At the same time, phenolic hydroxyl groups act as hydrophilic substances, making water easier to pass through the film [44]. We have revised the result according to your comments and added some reasons. Thank you very much!
Point 17: Line 232-233: The combination of RCP molecules and chitosan blocks the interaction between chitosan and glycerol, which reduces the TS of the composite film: NO, if we block the interaction between chitosan and glycerol the EAB is more affected. Be sure that the plasticizer affects more the EAB and YM and the addition of crosslinker affect more the TS. In addition, this sentence is in contradiction with the line 236: The effect of RCP extract is similar to that of a plasticizer and improves mechanical properties of the film: if mechanical properties is improved TS is improved. In your sentence you can say that RCP play a plasticizer role so the EAB increased. Pease add the SD to all value presented in the discussion part.
Revision: Page 8, line 279-283: The addition of RCP extract changes the molecular interaction between CS and GL, promotes the movement of molecules, destroyes the compact spatial structure of CS-GL film, and reduces the TS of the CS-GL-RCP films [45]. At the same time, the effect of RCP extract is similar to that of plasticizer, which plays a plasticizing role and improves the EAB of the CS-GL-RCP films [46]. We have revised the result according to your comments and added the SD to all value presented in the discussion part. Thank you very much!
Point 18: Line 244: Due to the presence of some hydroxyl groups in the anthocyanin in the RCP extract, the WCA decreases after adding the RCP extract [35]. Please explain more. In your case you should do the Critical surface tension and free tension energy with polar and dispersive compounds used 3 or 4 liquids.
Revision: Page 8, line 289-293: The hydrophilicity of the film helps to intelligently indicate the color change of the smart film in response to pH changes in the environment [48]. Compared with CS-GL film, the WCA of CS-GL-RCP films decreases with the increase of RCP extract content because the anthocyanins in RCP extract have a large number of highly absorbent hydroxyl groups, making CS-GL-RCP films more hydrophilic [39]. We have added some explains according to your comments. We think that the purpose of WCA is to increase the parameters of physicochemical characterization. The prepared CS-GL-RCP smart film is applied to pickle preservation, and it is not used as a food packaging film to package food. In the next experiment, We will add the Critical surface tension and free tension energy with polar and dispersive compounds used 3 or 4 liquids. Thank you very much!
Point 19: Line 274 the FTIR part should be revised: If we have some interaction between OH group of anthocyanin and NH3+ of chitosan you van get some shift relative to CH region.
Revision: Page 10, line 333-337: With the increase of the content of RCP extract in CS-GL-RCP films, the characteristic bands of CS-GL-RCP films have some displacement changes, which is due to the interaction between the hydroxyl group of the anthocyanin in the film and the amino group of the chitosan, which also confirms that the RCP extract is successfully fixed to the CS-GL film [51]. We have revised the result according to your comments. Thank you very much!
Point 20: In addition, in the figure of FTIR spectra please add the involved group of each peak. One example NH-amide II band 1540-1570; CO amide I in 1630-1660…..
Revision: Page 11, line 340: We have revised the figure of FTIR spectra according to your comments and added some reasons. Thank you very much!
Point 21: Conclusion
L323: should be more consistent and different to abstract. Add also the perspective.
Revision: Page 13-14, line 377-390: The CS-GL-RCP composite films were prepared by solvent evaporation. The results showed that RCP anthocyanins were successfully compounded onto CS-GL films and formed new interactions with CS-GL films. At different pH values, the color of the RCP extract is different and can be visually distinguished. With the addition of RCP extract, the physicochemical characterization and color parameters of the four films changed. The RCP extract increases the thickness and water vapor permeability (WVP) of the films. The RCP extract obviously reduces (p<0.05) water content (WC), swelling property (SP) and water contact angle (WAC) of the films. In addition, the CS-GL-RCP composite film has better flexibility and UV-Vis light barrier property than the CS-GL film. The CS-GL-RCP film is smart and its color changes obviously in the solution with different pH values. The color change of the film is in good agreement with the maturation time point of the pickle. CS-GL-RCP15 film is recommended as a smart film to monitor the fermentation process of pickles stored at 20±1 °C for 15 days and has broad application prospects in monitoring the maturity of pickle fermentation process. We have revised the conclusions according to your comments. Thank you very much!

Reviewer 3 Report (New Reviewer)
A study on “Development of smart films based on chitosan-glycerol and Robusta Coffee peel extract for monitoring the fermentation process of pickle” has been investigated. Many tests were conducted by the authors. The paper is well written and discussed. The manuscript, however, needs some improvement. I do suggest to improve the manuscript. The recommendations are the following:
It is not necessary to include glycerol in the title since it is a well-known plasticizer used for film preparation. I recommend changing the title to “Development of smart films of chitosan-based and Robusta Coffee peel extract for monitoring the fermentation process of a pickle.”
Abstract; “Therefore, the CS-GL-RCP film prepared in this study provided a new strategy for the development of active packaging materials” what the authors mean with this sentence. Do you mean prepare film provided a new strategy for development of pickle (fermentation process of pickle) by monitoring the pH? It is vague.
please add quantitative data to the abstract.
As an introduction, I would suggest starting with a gap of study, such as low food shelf life, spoilage of food and diseases caused by food microorganisms, as well as the importance of intelligent packaging in the food industry.
I recommend that to mention the gap and novelty of study at the end of introduction.
There is insufficient detail of the film preparation! What time and temperature were used for the preparation process? What temperature and RH were used for film maintenance?
Author Response
Response to reviewer’s comments
Manuscript ID: Foods-2341145
Title: Development of smart films based on chitosan-glycerol and Robusta Coffee peel extract for monitoring the fermentation process of pickle.
Journal: Foods.
Dear reviewer,
I have made revisions according to reviewer's comments. It had been appended below. The revised section was shown in red color.
Yours sincerely,
Yuyue Qin
Reviewer comments:
Reviewer: A study on “Development of smart films based on chitosan-glycerol and Robusta Coffee peel extract for monitoring the fermentation process of pickle” has been investigated. Many tests were conducted by the authors. The paper is well written and discussed. The manuscript, however, needs some improvement. I do suggest to improve the manuscript. The recommendations are the following:
Revision: We have revised the manuscript according to your comments. Thank you very much!
Specific comment:
Point 1: Title: It is not necessary to include glycerol in the title since it is a well-known plasticizer used for film preparation. I recommend changing the title to “Development of smart films of chitosan-based and Robusta Coffee peel extract for monitoring the fermentation process of a pickle.”
Revision: Page 1, line 2-4: The title“Development of smart films of chitosan-based and Robusta Coffee peel extract for monitoring the fermentation process of a pickle”has been modified. Thank you very much!
Point 2: Abstract:“Therefore, the CS-GL-RCP film prepared in this study provided a new strategy for the development of active packaging materials”what the authors mean with this sentence. Do you mean prepare film provided a new strategy for development of pickle (fermentation process of pickle) by monitoring the pH? It is vague.
Revision: Page 1, line 26-27: Therefore, CS-GL-RCP films prepared in this study provided a new strategy for the development of smart packaging materials. I have two meanings to explain this sentence. 1.The smart film was prepared by adding anthocyanin-rich Robusta coffee peel (RCP) extract into chitosan (CS)-glycerol (GL) matrix by solution casting method. The RCP extract is applied to the smart film to enrich the anthocyanins used in the smart packaging materials. 2. Other smart films are more applied to meat preservation. The change of color is due to the change of pH in the environment to alkaline. The CS-GL-RCP film is applied to the fermentation of pickles. The change of color is due to the change of pH in the environment to acidic, enriching the use of smart packaging materials. Thank you very much!
Point 3:please add quantitative data to the abstract.
Revision: Page 1, line 13-27: By changing the content of RCP (0, 10%, 15% and 20%) in CS-GL film, the related performance indicators of CS-GL-RCP films were studied. The results showed that the CS-GL-RCP films had excellent mechanical properties, and CS-GL-RCP-15 film maintained the excellent tensile strength (TS) of 16.69 MPa and elongation at break (EAB) of 18.68% with RCP extract. CS-GL-RCP films had the best UV-vis light barrier property at 200-350 nm and the UV transmittance was close to 0. The microstructure observation results showed that CS-GL-RCP films had a dense and uniform cross section, which proved that RCP extract had good compatibility with the polymer. In addition, CS-GL-RCP15 film was pH-sensitive and could exhibit different color changes with different pH solutions. So, the CS-GL-RCP15 film was used to detect the fermentation process of pickles at 20±1 °C for 15 days. The pickles were stored in a round pickle container after boiling water cooled. The color of CS-GL-RCP15 film changed significantly, which was consistent with the change of pickles from fresh to mature. The color of the smart film changed significantly with the maturity of pickles and the difference of ΔE of film increased to 8.89 (15 Day), which can be seen by the naked eye. Therefore, CS-GL-RCP films prepared in this study provided a new strategy for the development of smart packaging materials. The quantitative data has been added. Thank you very much!
Point 4: As an introduction, I would suggest starting with a gap of study, such as low food shelf life, spoilage of food and diseases caused by food microorganisms, as well as the importance of intelligent packaging in the food industry.
Revision: Page 2, line 32-40: Food has a low shelf life and is easily affected by endogenous enzymes and external microorganisms. In life, people's awareness of food safety is increasing year by year. With the increasing demand for food safety, how to better protect consumers from diseases caused by spoilage food and maximize food quality has become an important research direction. Therefore, people have a strong interest in advanced packaging materials. Smart packaging has risen rapidly and entered the sight of consumers. Smart food packaging can provide consumers with information, display changes in the food packaging environment and monitor the freshness of food in real time. It is of great significance to improve the food safety and reduce food waste [1]. We have revised the introduction according to your comments. Thank you very much!
Point 5: I recommend that to mention the gap and novelty of study at the end of introduction.
Revision: Page 2, line 67-73: The research on the preparation of smart film with Robusta Coffee peel (RCP) as raw material has not been carried out. At the same time, the previous smart films have been studied a lot in monitoring meat deterioration, but there are few reports on monitoring the maturity of pickles. The maturity of pickles is crucial to the safety of consumers. Therefore, RCP extract can be used as an intelligent pH additive to prepare smart films for pickle fermentation process and monitor the freshness change of pickles. The gap and novelty of study have been added according to your comments. Thank you very much!
Point 6: There is insufficient detail of the film preparation! What time and temperature were used for the preparation process? What temperature and RH were used for film maintenance?
Revision: Page 3, line 112-118: 2.0 g chitosan powder and 0.2 g glycerol were added in 100 mL acetic acid solution (2% v/v) and stirred for 12 h at 25 °C to get chitosan solution (2% w/v) [5]. Next, the RCP extract added to the mixed solution. The solution poured into apolytetrafluoroethylene plate and dried in an oven at 35 °C with a relative humidity of 50% for 12 h. When the content of RCP extract was 10%, 15% and 20% based on the weight of chitosan, the prepared films were named CS-GL, CS-GL-RCP10, CS-GL-RCP15 and CS-GL-RCP20, respectively. The insufficient detail of the film preparation has been added according to your comments. Thank you very much!

Round 2
Reviewer 1 Report (New Reviewer)
Improved after revision.
Author Response
Response to reviewer’s comments
Manuscript ID: Foods-2341145
Title: Development of smart films based on chitosan-glycerol and Robusta Coffee peel extract for monitoring the fermentation process of pickle.
Journal: Foods.
Dear reviewer,
Thank you very much!
Yours sincerely,
Yuyue Qin
Reviewer 2 Report (New Reviewer)
This paper dealing with the development of new smart films based on chitosan and anthocyanin-rich Robusta coffee peel extract. This paper gives an original data dealing with impact of RCP for monitoring the fermentation process of pickle. This revised version is more clear than the first version. But unfortunately the relationships between the reported properties of the extract and the interactions with chitosan must be more explained.The moisture content of the different film recipes should be more discussed. Very week explanation of the results. The manuscript is a series of analyses without any link between different parts. I very week discussion of results with the literature. A leak of explanation of plasticizer effect of RCP on films matrix, and it’s relation to structural and microstructural and barrier properties.The RH gradient for WVP is not explained and this parameter is very important: You said that you used sodium chloride at 95% RH at the bottom, yes I agree but the in the cup we can not considered at 0% RH: you should used silicagel or P2O5 to fixe the RH at around 0%. This paper need to be improved.
Author Response
Response to reviewer’s comments
Manuscript ID: Foods-2341145
Title: Development of smart films based on chitosan-glycerol and Robusta Coffee peel extract for monitoring the fermentation process of pickle.
Journal: Foods.
Dear reviewer,
I have made revisions according to reviewer's comments. It had been appended below. The revised section was shown in red color.
Yours sincerely,
Yuyue Qin
Reviewer comments:
Reviewer: This paper dealing with the development of new smart films based on chitosan and anthocyanin-rich Robusta coffee peel extract. This paper gives an original data dealing with impact of RCP for monitoring the fermentation process of pickle. This revised version is more clear than the first version.
Revision: We have revised the manuscript according to your comments. Thank you very much!
Specific comment:
Point 1: But unfortunately the relationships between the reported properties of the extract and the interactions with chitosan must be more explained.
Revision: In WC: Page 7, line 242-253: It is worth noting that the water content (WC) of CS-GL films is the highest, and the WC of CS-GL-RCP films is slightly reduced compared to CS-GL film. With the increased of anthocyanins content, the WC of films declines obviously (p<0.05) from 19.92%±0.55% to 11.09%±0.85%. The main reason for the formation of water content is the molecular interaction between water molecules and polymer [34]. The anthocyanins in the RCP extract form hydrogen bonds with the hydrophilic groups in chitosan, which promotes the compatibility of anthocyanins with the film [38]. Anthocyanins limited the interaction between chitosan and water molecules, reduced the interaction between hydroxyl and amino groups, produced a dense and continuous structure of film, and formed the CS-GL-RCP film with lower WC [39]. Peralta et al. reported that the addition of Aqueous hibiscus extract promoted a significant decrease in the WC of CS composite films (p<0.05) [40].
40.Peralta, J.; Bitencourt-Cervi, C.M.; Maciel, V.B.V.; Yoshida, C.M.P.; Carvalho, R.A. Aqueous hibiscus extract as a potential natural pH indicator incorporated in natural polymeric films. Food Packaging Shelf. 2019, 47-55.
In SP: Page 7, line 254-262: The SP of the film is showed in Table 1. The CS-GL-RCP20 film has the lowest swelling ratio (185%±0.78%), and the CS-GL film has the highest swelling ratio (202%±1.28%). The SP of the CS-GL-RCP composite films become lower because of the interaction between RCP extract and chitosan molecules [41]. The swelling ratio represents the efficiency of the film in response to color. The high swelling ratio leads to the rapid release of anthocyanin solution, which is not conducive to the visual observation of the color change of smart indication packaging film with the changes of pH value [42]. Therefore, the CS-GL-RCP films with low swelling ratio are used for food packaging materials.
In WVP: Page 7, line 263-274: The function of food packaging is to reduce the water exchange between air and food. WVP represents the water vapor permeability of the film, so the lower the WVP value of the film, the better the waterproof performance of the film [43]. The WVP of composite films is shown in Table 1. After adding low content of RCP extract (10%), the WVP does not increase obviously (p>0.05), but when content is 15%, the WVP increases obviously (p<0.05). Because the number of the RCP extract (10%) is limited, the impact on WVP of the composite film can be ignored. The anthocyanins in RCP extract contain a variety of phenolic hydroxyl hydrophilic substances, which change the original molecular interaction between CS and GL, and interfere with the dense structure of the film to a certain extent. At the same time, phenolic hydroxyl groups act as hydrophilic substances, making water easier to pass through the film [44]. The same change was reported in the article of black rice bran anthocyanins in chitosan/oxidized films [45].
In the physicochemical characterization of the three films, the relationship between extract properties and chitosan interaction has been introduced. At the same time, references with the same changes have been added to assist interpretation.
In microstructure: Page 9, line 314-325: The cross-section is shown in Figure 4. The CS-GL film is shown the smoothest cross section and no wrinkles. With addition of RCP extract, the CS-GL-RCP films show typical images with porous structure [45]. The cross section of CS-GL-RCP films is affected by addition of RCP extract. This is due to the formation of new hydrogen bonds between anthocyanin molecules and polymer chains. The CS-GL-RCP films show a tight structure in the cross-section diagram, indicating that the polymer and RCP extract had good compatibility and mixed solubility [41]. At the same time, the results confirm that RCP extract is evenly distributed on the films. In addition, the CS-GL-RCP20 film is rougher than other films and shows slight agglomeration. This may be because when the RCP extract reaches a certain high concentration, the solubility of the extract in the film-forming solution is limited. The results of cross section are consistent with the physical properties.
In FTIR: Page 10, line 330-345: As shown in Figure 5, the intermolecular interactions are examined by FTIR. The composite films have the wide band from 3395 to 3378 cm-1 (CH, alkyne stretching), 2961 to 2925 cm-1 (CH, alkane stretching), 1636 to 1606 cm-1 (C=C stretching), 1480 to 1460 cm−1 (C-H bend (in-plane), alkane), 1341 to 1300 cm−1 (C-H bend (in-plane), alkene), 1169 to 1129 cm−1 (C-C stretching), and 751 to 725 cm−1 (C-H bend (out of plane)). Because chitosan is a hydrophilic substance and the anthocyanins in RCP extract have hydroxyl group, the wide band from 3395 to 3378 cm-1 (CH, alkyne stretching) of the CS-GL-RCP films observes is a hydrophilic substance. When the material is mixed, the physical mixing leads to the change of the spectral peak [50]. With the increase of the content of RCP extract in CS-GL-RCP films, the characteristic bands of CS-GL-RCP films have some displacement changes, which is due to the interaction between the hydroxyl group of the anthocyanin in the films and the amino group of the chitosan, which also confirms that the RCP extract is successfully fixed to the CS-GL film [51]. Many researchers had proposed that there was a certain interaction between the film component and the band [50]. This is due to the formation of hydrogen bonds between the amino and hydroxyl groups in the film and the hydroxyl groups in the RCP extract.
The relationship between the properties of the extract and the interaction of chitosan was also modified by SEM and FTIR according to your tips. Thank you very much!
Point 2: The moisture content of the different film recipes should be more discussed.
Revision: Page 7, line 242-253: It is worth noting that the water content (WC) of CS-GL films is the highest, and the WC of CS-GL-RCP films is slightly reduced compared to CS-GL film. With the increased of anthocyanins content, the WC of films declines obviously (p<0.05) from 19.92%±0.55% to 11.09%±0.85%. The main reason for the formation of water content is the molecular interaction between water molecules and polymer [34]. The anthocyanins in the RCP extract form hydrogen bonds with the hydrophilic groups in chitosan, which promotes the compatibility of anthocyanins with the film [38]. Anthocyanins limited the interaction between chitosan and water molecules, reduced the interaction between hydroxyl and amino groups, produced a dense and continuous structure of film, and formed the CS-GL-RCP film with lower WC [39]. Peralta et al. reported that the addition of Aqueous hibiscus extract promoted a significant decrease in the WC of CS composite films (p<0.05) [40].
40.Peralta, J.; Bitencourt-Cervi, C.M.; Maciel, V.B.V.; Yoshida, C.M.P.; Carvalho, R.A. Aqueous hibiscus extract as a potential natural pH indicator incorporated in natural polymeric films. Food Packaging Shelf. 2019, 47-55.
The moisture content part has been extended according to your opinion. Thank you very much!
Point 3: Very week explanation of the results. The manuscript is a series of analyses without any link between different parts. I very week discussion of results with the literature.
Revision: Page 8, line 294-296: The changes of WCA and WVP of the films are consistent, which can synergistically explain the related changes. Page 9, line 324-325: The results of cross section are consistent with the physical properties. Page 9, line 338-345: With the increase of the content of RCP extract in CS-GL-RCP films, the characteristic bands of CS-GL-RCP films have some displacement changes, which is due to the interaction between the hydroxyl group of the anthocyanin in the films and the amino group of the chitosan, which also confirms that the RCP extract is successfully fixed to the CS-GL film [51]. Many researchers had proposed that there was a certain interaction between the film component and the band [50]. This is due to the formation of hydrogen bonds between the amino and hydroxyl groups in the film and the hydroxyl groups in the RCP extract. Following your prompts, links between different parts have been added. Thank you very much!
Point 4: A leak of explanation of plasticizer effect of RCP on films matrix, and it’s relation to structural and microstructural and barrier properties.
Revision: In mechanical properties: Page 8, line 283-284: At the same time, the effect of RCP extract is similar to that of plasticizer, which plays a plasticizing role and improves the EAB of the CS-GL-RCP films [46].
It’s relation to structural and microstructural: In microstructure: Page 9, line 314-325: The cross-section is shown in Figure 4. The CS-GL film is shown the smoothest cross section and no wrinkles. With addition of RCP extract, the CS-GL-RCP films show typical images with porous structure [45]. The cross section of CS-GL-RCP films is affected by addition of RCP extract. This is due to the formation of new hydrogen bonds between anthocyanin molecules and polymer chains. The CS-GL-RCP films show a tight structure in the cross-section diagram, indicating that the polymer and RCP extract had good compatibility and mixed solubility [41]. At the same time, the results confirm that RCP extract is evenly distributed on the films. In addition, the CS-GL-RCP20 film is rougher than other films and shows slight agglomeration. This may be because when the RCP extract reaches a certain high concentration, the solubility of the extract in the film-forming solution is limited. The results of cross section are consistent with the physical properties.
In FTIR: Page 10, line 330-345: As shown in Figure 5, the intermolecular interactions are examined by FTIR. The composite films have the wide band from 3395 to 3378 cm-1 (CH, alkyne stretching), 2961 to 2925 cm-1 (CH, alkane stretching), 1636 to 1606 cm-1 (C=C stretching), 1480 to 1460 cm−1 (C-H bend (in-plane), alkane), 1341 to 1300 cm−1 (C-H bend (in-plane), alkene), 1169 to 1129 cm−1 (C-C stretching), and 751 to 725 cm−1 (C-H bend (out of plane)). Because chitosan is a hydrophilic substance and the anthocyanins in RCP extract have hydroxyl group, the wide band from 3395 to 3378 cm-1 (CH, alkyne stretching) of the CS-GL-RCP films observes is a hydrophilic substance. When the material is mixed, the physical mixing leads to the change of the spectral peak [50]. With the increase of the content of RCP extract in CS-GL-RCP films, the characteristic bands of CS-GL-RCP films have some displacement changes, which is due to the interaction between the hydroxyl group of the anthocyanin in the films and the amino group of the chitosan, which also confirms that the RCP extract is successfully fixed to the CS-GL film [51]. Many researchers had proposed that there was a certain interaction between the film component and the band [50]. This is due to the formation of hydrogen bonds between the amino and hydroxyl groups in the film and the hydroxyl groups in the RCP extract.
It’s barrier properties: Page 8, line 304-310: Because the CS-GL-RCP compisite film contains the anthocyanins, anthocyanins have the potential to absorb ultraviolet light [48]. Ultraviolet rays increase the scattering reaction and refraction reaction when passing through the CS-GL-RCP compisite film, so that the ultraviolet transmittance of the film is greatly reduced, which is very important to protect the quality of food in the package [45]. Therefore, the CS-GL-RCP film can be used for food packaging because of its excellent light barrier. We modified the explanation of plasticizer effect of RCP according to your opinion. Thank you very much!
Point 5: The RH gradient for WVP is not explained and this parameter is very important: You said that you used sodium chloride at 95% RH at the bottom, yes I agree but the in the cup we can not considered at 0% RH: you should used silicagel or P2O5 to fixe the RH at around 0%. This paper need to be improved.
Revision: Page 4, line 140-147: 5.0 g silica gel was put into the weighting test cup (relative humidity 0%). The cup was covered with the film and placed in a glass vacuum desiccator equipped with saturated sodium chloride solution (25 ℃, relative humidity 95%) at the bottom. Due to the presence of saturated sodium chloride solution, the vapor pressure in the cup is considered to be zero. The external vapor pressure of the film is also obtained by the product of the relative humidity (95%) in the glass vacuum desiccator and the pure water vapor pressure at 25 ℃. Therefore, a certain vapor pressure difference is maintained on both sides of the film. We modified the material part according to your opinion. Before, there was a writing error. The silica gel was used in the experiment. The WVP tested by weighing method according ASTM E96 standard and this paper [24]. Thank you very much!

Round 3
Reviewer 2 Report (New Reviewer)
No further comments.
Author Response
Response to reviewer’s comments
Manuscript ID: Foods-2341145
Title: Development of smart films based on chitosan-glycerol and Robusta Coffee peel extract for monitoring the fermentation process of pickle.
Journal: Foods.
Dear reviewer,
Thank you very much!
Yours sincerely,
Yuyue Qin
This manuscript is a resubmission of an earlier submission. The following is a list of the peer review reports and author responses from that submission.
Round 1
Reviewer 1 Report
In the current manuscript, the preparation of smart films based on chitosan-glycerol and Coffee peel extract to monitor the fermentation process of pickle were studied. In my opinion, the manuscript has a good structure and has innovation and appropriate tests.
L20: Write the suggestion (recommendation) of the findings of this research in one sentence at the end of the abstract.
L30: "a pH" is correct.
L35: Add reference for this sentence: "The CS film has low mechanical strength".
L39-41: Add reference/s for these lines.
L43-45: In the introduction, all sentences should contain appropriate references.
L46: write "Phyllanthus reticulatus" in italic form.
L 67: How did you prepare the coffee peel? Describe.
L69: power of ultrasonication?
L72: write more details about the freezing and drying.
L94: write the specifics of dryer.
L138: write more details about the color changes.
-please put * above the a and b in color parameters such as a*.
-did the addition of the RCP extract affect the FTIR spectrum?
Author Response
We have revised the manuscript according to your comments. Thank you very much!

Author Response

(The authors gave the same response as above.)
